# Nuclear Localization of PTTG1 Promotes Migration and Invasion of Seminoma Tumor through Activation of MMP-2

**DOI:** 10.3390/cancers13020212

**Published:** 2021-01-08

**Authors:** Emanuela Teveroni, Fiorella Di Nicuolo, Giada Bianchetti, Alan L. Epstein, Giuseppe Grande, Giuseppe Maulucci, Marco De Spirito, Alfredo Pontecorvi, Domenico Milardi, Francesca Mancini

**Affiliations:** 1International Scientific Institute “Paul VI”, ISI, Fondazione Policlinico ‘A. Gemelli’ IRCCS, 00100 Rome, Italy; ema.teveroni@gmail.com (E.T.); fiorella.dinicuolo@gmail.com (F.D.N.); grandegius@gmail.com (G.G.); alfredo.pontecorvi@unicatt.it (A.P.); milardid@yahoo.it (D.M.); 2Department of Neuroscience, Section of Biophysics, Università Cattolica del Sacro Cuore, 00100 Roma, Italy; giada.bianchetti@unicatt.it (G.B.); giuseppe.maulucci@unicatt.it (G.M.); marco.despirito@unicatt.it (M.D.S.); 3Department of Pathology, Keck School of Medicine, University of Southern California, Los Angeles, CA 90001, USA; aepstein@med.usc.edu; 4Division of Endocrinology, Fondazione Policlinico ‘A. Gemelli’ IRCCS, 00100 Rome, Italy

**Keywords:** PTTG1, seminoma, testicular cancer, MMP-2, invasiveness

## Abstract

**Simple Summary:**

Seminoma is the most common subtype of testicular germ cell tumors (TGCTs) and its molecular patterns have not been clarified. The pituitary tumor-transforming gene 1 (PTTG1) is a securin and its overexpression is reported in many cancers. We previously demonstrated that PTTG1 is mainly localized at the neoplasm periphery and infiltration area of seminoma. Therefore, we aim to investigate in vitro the role of PTTG1 on the invasive properties of seminoma. Our results elucidate the role of nuclear PTTG1 in promoting invasiveness and the metastatic process of these cells through its transcriptional target matrix-metalloproteinase-2 (MMP-2). Analysis of human testicular tumors from the Atlas database revealed an exclusive PTTG1 nuclear localization and a concomitant increase of MMP-2 levels in seminoma compared to non-seminoma tumors. Our data provide insights into the molecular characterization of seminoma, promoting PTTG1 as a prognostic marker useful in human seminoma clinical management.

**Abstract:**

(1) Background: PTTG1 sustains the invasiveness of several cancer types. We previously reported that in seminomas, PTTG1 was detected in the peripheral area of the tumor and in the leading infiltrative edge. Here, we investigate the PTTG1 role on the invasive properties of seminoma. (2) Methods: three seminoma cell lines were used as in vitro model. PTTG1 levels and localization were investigated by biochemical and immunofluorescence analyses. Wound-healing, Matrigel invasion assays, and zymography were applied to study migratory and invasive capability of the cell lines. RNA interference and overexpression experiments were performed to address the PTTG1 role in seminoma invasiveness. PTTG1 and its target MMP-2 were analyzed in human testicular tumors using the Atlas database. (3) Results: PTTG1 was highly and differentially expressed in the seminoma cell lines. Nuclear PTTG1 was positively correlated to the aggressive phenotype. Its modulation confirms these results. Atlas database analysis revealed that PTTG1 was localized in the nucleus in seminoma compared with non-seminoma tumors, and that MMP-2 levels were significantly higher in seminomas. (4) Conclusions: nuclear PTTG1 promotes invasiveness of seminoma cell lines. Atlas database supported these results. These data lead to the hypothesis that nuclear PTTG1 is an eligible prognostic factor in seminomas.

## 1. Introduction

Seminoma is the most common histological subtype of testicular germ cell tumors (TGCTs), and accounts for 1% of all cancers in men. It is most frequent in men 20–40 years old [1]. The pre-invasive stage of carcinoma in situ (CIS) subsumes all TGCTs which can then develop into non-seminomas or into seminoma cancer [2]. Despite the etiology of TGCTs being well studied, their molecular patterns and regulatory mechanisms have not been thoroughly investigated [3].

The pituitary tumor-transforming gene 1 (PTTG1) is a securin, inhibitor of premature sister chromatid segregation during mitosis process [4]. It also possesses transcriptional activity on several targets involved in different cellular processes such as proliferation, angiogenesis, and invasion [5].

PTTG1 overexpression is reported to exert its oncogenic function by altering sister chromatid separation during cell division, leading to aneuploidy [6,7]. In normal tissues, PTTG1 is expressed at low levels with the only exception of testis, in which it is reported relatively higher than other tissues, but lower than pituitary adenomas [8]. In fact, PTTG1 overexpression is reported in many cancer types, such as pituitary, thyroid [9,10,11,12], brain [13,14], and breast [15]. As an oncogene, overexpressed PTTG1 causes aneuploidy and genetic instability and its contribution to tumorigenesis is also accountable to the promotion of invasiveness by its transcriptional activity. Indeed, PTTG1 contributes to the metastatic process and tumor progression by transactivation of matrix metalloproteinase-2 (MMP-2) [16].

Previously, we analyzed PTTG1 localization in a subset of human testicular cancers [17]. Immunostaining analysis revealed that in in situ testicular cancer (CIS), PTTG1 showed nuclear staining only in isolated cells. Of interest, in seminomas, PTTG1 was localized in the cytoplasm in the central area of the tumor while in the periphery it was mainly localized in the nuclei, leading us to hypothesize that the more invasive-prone cancer area requires a nuclear localization of this securin [17]. Accordingly, it has been reported that PTTG1 cytoplasmic localization was more frequent in normal tissues and in pituitary adenomas [18,19,20], while the nuclear localization was associated with a more aggressive phenotype [21] and to tumor recurrence [22].

The PTTG1 interacting partner, named pituitary tumor-transforming gene (PTTG)-binding factor (PBF), directly interacts with PTTG1, promotes the shift of securin from the cell cytoplasm to the nucleus [23], and was found to be involved in the progression of different tumor types [24,25,26]. Indeed, PBF overexpression was closely associated with the clinical features of patients, including tumor recurrence, metastasis, and patients’ overall survival [27,28].

The aim of the present work was to investigate the role of PTTG1 in seminoma cancer progression. In particular, we evaluated the role of PTTG1 nuclear fraction in human seminoma-derived cell cultures. We made use of three different seminoma cell lines, characterized for their proliferation rate, cytogenetic, and marker expression features [29]. The TCAM2 cell line was derived from a human seminoma established by Mizuno [30], the JKT-1 cell line was established from a primary lesion of a left testicular seminoma [31], and the SEM-1 was established from an extragonadal seminoma, obtained from anterior mediastinal mass [29].

We found that PTTG1 expression levels and in particular its nuclear localization strongly correlated with the invasive properties of the three different cell lines.

PTTG1’s role in promoting an aggressive phenotype is supported by RNA interference experiments in JKT-1 and SEM-1, showing a significant reduction in invasion and MMP-2 activity. Moreover, PTTG1 overexpression in TCAM2 showed an increase in the MMP-2 protein level. Of note, TCAM2, which showed the lowest level of nuclear PTTG1, was relatively resistant to PTTG1 nuclear translocation, despite the exogenous overexpression of PTTG1 or its nuclear localizing factor PBF, suggesting other underlying factors responsible for this behavior.

Of interest, analysis of the Atlas database of testicular tumors supported in vivo the role of PTTG1 in seminoma [32]. A database interrogation revealed that exclusively in seminoma PTTG1 was localized in the nucleus compared with non-seminoma tumors. The analysis demonstrates a significantly higher level of MMP-2 in seminomas compared to non-seminoma tumors, supporting the role of PTTG1 nuclear activity in driving MMP-2 levels and hence in promoting invasiveness of seminoma tumors.

Taken together, these findings support the role of PTTG1 nuclear localization in promoting invasiveness and metastasis, leading to the hypothesis that nuclear PTTG1 eligibility is a potential prognostic factor for seminomas.

## 2. Results

### 2.1. Analysis of PTTG1 Protein Levels and Its Subcellular Localization in Seminoma Cell Lines

Since we previously reported a PTTG1 nuclear expression in the peripheral area of human seminoma, we wondered if the securin subcellular localization is involved in the progression of these tumors. To this aim, we initially evaluated PTTG1 protein levels in the three cell lines that showed different growth properties [29]. Immunoblot analysis of whole cell extracts show that PTTG1 was overexpressed in the seminoma cell lines compared to PC3 cells, prostatic adenocarcinoma derived cells in which PTTG1 is reported to be upregulated [33,34] (Figure 1A,B). Intriguingly, PTTG1 levels were different between the cell lines with TCAM2 showing the lowest expression (Figure 1A,B).

We then analyzed PTTG1 subcellular localization. In all the cell lines, PTTG1 was more cytoplasmic than nuclear (Figure 1C). In particular, in TCAM2 cells, PTTG1 nuclear fraction was the lowest in comparison to JKT-1 and SEM-1 cells (Figure 1C). Moreover, TCAM2 cells showed the lowest nuclear/cytoplasmic ratio (Figure 1D).

Confocal PTTG1 immunofluorescence analysis confirmed these results (Figure 1E). The co-localization rate was analyzed by evaluating the threshold of Manders’ co-localization coefficients (tM2 coefficient), which represents the fraction of PTTG1 protein co-localized with DAPI nuclear staining. The tM2 was 10% for TCAM2 while it was significantly higher, around 50%, for JKT-1 and SEM-1 (Figure 1F).

These data demonstrate that PTTG1 is highly expressed in all the three seminoma cell lines, and it is differentially expressed and localized.

### 2.2. Migratory and Invasive Properties of Seminoma Cell Lines

We wondered whether differential PTTG1 expression was correlated to the invasive properties of the three cell lines. To address this point we performed functional assays. At first, we evaluated the migratory capability of seminoma cell lines using a wound-healing assay. As reported in Figure 2A,B in JKT-1 and SEM-1 the gap length was significantly reduced after 4 h, with a tendency that lasted over 24 h. Conversely, in TCAM2 cells, the gap remained almost the same and its shrinkage was not statistically significant after 24 h (Figure 2A,B).

The cell migratory property should be accompanied by substrate invasion ability to define a more reliable malignancy pattern in cancer cell lines. To this aim, we made use of Matrigel-coated transwell chambers assay. We measured the invasion index seen as the percentage of cells that crossed the Matrigel compared to the total number of seeded cells at time zero. After 48 h, all three seminoma cell lines showed significant differences in their invasion index (Figure 2C). In particular, JKT-1, SEM-1, and TCAM2 had a 40%, 25%, and 10% invasion index, respectively (Figure 2C). Of note, the progressive decrease of invasion percentage between the cell lines correlated to PTTG1 levels (Figure 1A,C).

Since it has been reported that PTTG1 enhances MMP-2 expression and activity [16], we investigated metalloproteinases secretion/activity in the seminoma cell lines by zymography. As shown in Figure 2D (and Appendix A), in JKT-1 and SEM-1, active MMP-2 was significantly higher compared to TCAM2, in which MMP-2 was barely detectable. Moreover, MMP-2 protein levels showed a similar trend compared to its activity, although one does not exactly reflect the other (Figure 2E,F).

Taken together, these data indicate that PTTG1 levels correlated with the migratory and invasive properties of the seminoma cell lines, supporting the hypothesis that PTTG1 protein levels and more interestingly its nuclear fraction are important for cancer progression and metastatic properties in seminoma cells.

### 2.3. Role of PTTG1 in Seminoma Cell Invasiveness

To uncover a causal link between PTTG1 and seminoma progression, we modulated the securin levels in the three different cell lines. Since JKT-1 and SEM-1 showed high PTTG1 levels, we downregulated the securin by siRNA (small interference RNA) in these cells. The RNA interference of PTTG1 almost abrogated its protein levels (Figure 3A,B) and this in turn significantly reduced the invasion capability of these cell lines (Figure 3C). Additionally, a decreased PTTG1 amount correlated with a significant reduction of MMP-2 activity (Figure 3D and Appendix A). Conversely, we overexpressed PTTG1 in TCAM2 cells. The subcellular fractionation analysis revealed that although the high cytoplasmic overexpression of FLAG-PTTG1 (Figure 3E) or pcDNA3.1-PTTG1 (Appendix A), the nuclear fraction of the protein was very low. We then analyzed MMP-2 levels and activity upon PTTG1 overexpression in TCAM2 cells and found that the overexpression was able to slightly increase the MMP-2 protein level (Figure 3F,G) but this was insufficient to induce MMP-2 secretion and activation in the zymography experiments (Appendix A). The failure in the rescue of a more aggressive phenotype in TCAM2 cells upon PTTG1 overexpression seems to be contradictory but it can be explained by the poor PTTG1 nuclear localization. As a control, we overexpressed both vectors carrying PTTG1 in JKT-1 cells (Appendix A). Western blot analysis of the cytoplasmic/nuclear fractions revealed that PTTG1 was overexpressed with both plasmids in the cytoplasm and in the nuclei with similar efficiency (Appendix A).

These results highlight the role of PTTG1 nuclear localization in promoting invasiveness of the seminoma cell lines.

### 2.4. Involvement of PTTG1 Binding Factor (PBF) in PTTG1 Subcellular Localization in Seminoma Cells

Since PBF is reported to mediate PTTG1 nuclear translocation [23], we wondered if this protein was involved in PTTG1 differential localization in three seminoma cell lines. At first, we analyzed the PBF protein level in whole cell extracts. PBF levels were similar between the three cell lines (Figure 4A,B left panel). Notably, TCAM2 cells showed the highest PBF/PTTG1 ratio, despite low PTTG1 nuclear level (Figure 4B right panel), suggesting that the differential behavior between the cell lines was independent of PBF protein levels. This prompted us to investigate the co-localization of PBF and PTTG1 in the cell lines. Confocal microscopy analysis revealed that in TCAM2, the co-localization index between the two proteins was significantly lower with respect to other cell lines (Figure 4C), with values of tM2 coefficient ranging from approximately 50% in TCAM2 to ~80% in JKT-1 and SEM-1 cells (Figure 4D). These data could explain, at least in part, the poor PTTG1 nuclear localization in TCAM2 cells.

Overexpression of PBF in TCAM2 cells and in control JKT-1 cells, analyzed by Western blot of cell fractions, showed that PBF overexpression caused partial nuclear translocation of PTTG1 only in JKT-1 cells, with TCAM2 being resistant to this phenomenon (Figure 4E,F). These results indicate the presence of a specific underlying mechanism responsible for PTTG1 cytoplasmic retention in TCAM2.

Overall, these results suggest that other players beyond PBF could determine nuclear PTTG1 localization in some seminoma tumors.

### 2.5. Analysis of PTTG1/PBF/MMP-2 Players in Seminoma Tumors

In order to validate in vivo the role of PTTG1 in seminoma, we wondered whether PTTG1, PBF, and MMP-2 showed specific behavior in human testicular tumors.

Using the Atlas database [32], first we analyzed PTTG1 RNA levels in seminoma (S) versus non-seminomas (N-S) testicular tumors and showed that PTTG1 levels were significantly lower in seminomas (Figure 5A). Since in our in vitro model we observed that the nuclear PTTG1 fraction drives the invasive properties of the cells, we investigated PTTG1 subcellular localization between the two groups using Atlas immunohistochemistry data (Figure 5B). Interestingly, PTTG1 showed a cytoplasmic/membranous/nuclear localization (C/N) in all seminoma specimens while in non-seminoma samples, it mainly localized in the cytoplasmic/membranous compartment (C) (Figure 5B). These results support the hypothesis that PTTG1 nuclear localization was a specific feature of the seminoma histotype among testicular tumors.

Since PBF mediates PTTG1 nuclear translocation [23], we would expect higher PBF levels in seminoma specimens compared to non-seminoma. On the contrary, PBF levels were significantly lower in seminomas (Figure 5C). Moreover, PBF was localized in the cytoplasmic/membranous compartment (C) in both categories (Figure 5D). These results suggest that PBF levels are not responsible for the PTTG1 nuclear localization in seminoma tumors.

Previously, we found that PTTG1 nuclear localization was strongly correlated with MMP-2 activity in our seminoma in vitro model. An interrogation of the same Atlas database, to evaluate MMP-2 levels in seminoma compared with non-seminoma tumors, showed a significant higher level of MMP-2 in seminoma samples (Figure 5E), supporting the role of PTTG1 nuclear transcriptional activity in driving MMP-2 levels. These data support the role of nuclear PTTG1 in promoting invasiveness of seminoma tumors.

## 3. Discussion

The results of the present research highlight the role of nuclear PTTG1 fraction on the invasive properties of human seminoma tumors.

PTTG1 is widely known as an oncogene involved in the development of several cancers [6,9,13,15]. In a recent gene ontology (GO) study that compared seminomas with normal testis tissues, a general dysregulation of many genes involved in cell adhesion and in cancer progression was reported [35]. Interestingly these investigators found a modulation of PTTG1 and MMP-2 expression [35]. The exact biological effects of PTTG1 on testis cancer carcinogenesis and progression remain unclear and its functional role has not yet been fully explored.

We previously evaluated the securin expression in different histological subtypes of testicular tumors by immunohistochemistry [17]. Above all, in seminomas, we reported a differential PTTG1 localization. In the central area of the tumor, PTTG1 staining was more intense in the cytoplasm whereas in the peripheral area, PTTG1 was mostly detected in the nucleus [17]. Interestingly, PTTG1-positive cells were also present in the leading infiltrative edge of the seminomas [17]. Moreover, we showed that the PTTG1 positive cell population in peripheral areas was characterized as octamer-binding transcription factor 4 (OCT4) and transcription factor Krüppel-like factor 4 (KLF4) positive [36], proteins associated with cancer stem cell self-renewal and invasiveness.

These results prompted us to investigate further the role of PTTG1 levels and cellular localization in seminoma progression. To this aim, we made use of three seminoma cell lines, representing unique models in the literature to study seminoma features in vitro. TCAM2 and JKT-1 cells derived from human primary seminomas, while SEM-1 cells derived from an extragonadal seminoma; all cell lines were characterized for growth parameters, morphology, and biomarker expression [29].

In the current study, we now demonstrate that PTTG1 is highly expressed in these three seminoma cell lines. Of interest, PTTG1 protein is differentially expressed in the seminoma cell lines, with TCAM2 cells presenting the lowest levels. These results show that PTTG1 is overexpressed in human seminoma.

It has been reported that only nuclear PTTG1 expression is associated with an aggressive phenotype, at least in pituitary tumors [21]. In our experiments, subcellular fractionation and immunostaining revealed that PTTG1 was localized both in the cytoplasm and in the nucleus of JKT-1 and SEM-1, while it was localized mainly in the cytoplasmic fraction of TCAM2 cells. Accordingly, migration and invasion assays as well as MMP-2 levels and activity were correlated with nuclear PTTG1 localization in the three different cell lines, with TCAM2 showing the lowest invasive capabilities. The role of nuclear PTTG1 in promoting an aggressive phenotype is supported by our RNA interference experiments in JKT-1 and SEM-1 that showed a significant reduction in invasion and MMP-2 activity. On the contrary, PTTG1 overexpression in TCAM2 was able to increase the MMP-2 protein level, but it was not sufficient to increase MMP-2 activity, due to the very poor nuclear overexpression despite a high transfection efficiency.

The main PTTG1 interacting protein, PBF, was reported to be responsible for PTTG1 nuclear translocation [23] and this function correlated with its oncogenic activity [27].

In our in vitro model, in TCAM2 cells, despite PBF overexpression, PTTG1 was not able to efficiently translocate into the nucleus. On the contrary, in JKT-1 cells PBF overexpression mediated, at least in part, PTTG1 nuclear relocalization. These results lead to the hypothesis that in TCAM2 cells, PTTG1 is sequestered in the cytoplasm by specific interactors and/or by post-translational modifications (PTMs) that impair PTTG1 interaction with PBF. For instance, Mora-Santos et al. demonstrated that a specific phosphorylation of PTTG1 was responsible for its nuclear localization [37]. Moreover, other researchers reported that Cyclin-dependent Kinase 1 (CDK1) mediated PTTG1 phosphorylation impairs its Golgi membrane localization [38]. Taken together, these data support the hypothesis that specific PTTG1 PTMs could be responsible for its subcellular localization. Ongoing studies are focused on the identification of PTTG1 PTMs or cytoplasmic interactors that inhibit its nuclear translocation in order to uncover new prognostic/therapeutic factors useful in the clinical management of seminomas.

Of interest, we validated the in vivo role of nuclear PTTG1 in seminoma via interrogation of the Atlas database of human testicular cancer [32]. This analysis revealed that PTTG1 was localized in the nucleus exclusively in seminoma, despite its lower levels in this group, compared to non-seminoma tumors. These data support the hypothesis that nuclear PTTG1 was a specific feature of seminoma compared to others testicular tumors.

Moreover, analysis of the Atlas database revealed that PBF levels were lower in seminomas compared to non-seminoma tumors, supporting the hypothesis that other players could determine nuclear PTTG1 localization in seminoma.

Importantly, this analysis further demonstrated a significant higher level of MMP-2 in seminomas compared to non-seminoma tumors, supporting the role of PTTG1 nuclear activity in driving MMP-2 levels and hence in promoting invasiveness of these tumors.

Overall, PTTG1 subcellular localization seems to be a significant factor in determination of its oncogenic role. It is tempting to speculate that the three cell lines, bearing different PTTG1 nuclear levels, could resemble distinct stages of seminoma. The identification of factors responsible for progressive PTTG1 nuclear translocation will help to elucidate seminoma cancer biology and to uncover new players useful in seminoma behavior and prognosis.

## 4. Materials and Methods

### 4.1. Cell Culture and Transfections

The three cell lines kindly provided by Dr. Epstein were cultured in RPMI supplemented with 10% FBS (Millipore, Burlington, MA, USA) and stable glutamine (Glutamax, Thermo Fisher, Waltham, MA, USA). Transient transfections were performed using Get Prime Polyplus according to manufacturer’s instructions (Polyplus, Illkirch-Graffenstaden, France). Plasmids used were: pcDNA3.1, pCMV (as control vectors -CTR), pcDNA3.1-PTTG1, pCMV-FLAG-PTTG1, and pCIneo HA-PBF.

PTTG1 siRNA and control siRNA were supplied by Invitrogen (Stealth RNAi) as a mix of three different siRNA. Cells were transfected using RNAiMAX reagent according to manufacturer’s instructions (Thermo Fisher, Waltham, MA, USA).

### 4.2. Immunofluorescence

Cells are fixed with 3.7% formaldehyde for 15 min at room temperature (RT), permeabilized with 0.05 Triton X-100 in PBS, and blocked with 5% bovine serum albumin (BSA). Cells were incubated with primary antibodies (rabbit α-PTTG1, Abcam, Cambridge, UK; mouse α-PBF, Novus Biologicals, Centennial, CO, USA) and then incubated with goat Alexa Fluo-488 anti-rabbit IgG and/or goat Alexa Fluo-594 anti-mouse IgG (Molecular Probes, Thermo Fisher, Waltham, MA, USA). DNA was stained with Prolong Gold DAPI (Molecular Probes).

### 4.3. Confocal Microscopy

Cells were imaged with an inverted confocal microscope (Nikon A1-MP). Fluorescence images (excitation: 402 nm for the blue channel and 488 nm for green channel) were collected in two separated channels (emission filter: 450/50 nm for the blue channel, 525/50 nm for the green channel) using a 60× immersion-oil objective with 1.4 numerical aperture (NA). Internal photon multiplier tubes collected 2048 × 2048-pixel images in 16-bit at 0.063 ms dwell time.

### 4.4. Western Blot Analysis

For Western blot, cells were lyzed in RIPA buffer (50 mM Tris–Cl, pH 7.5, 150 mM NaCl, 1% Nonidet P-40, 0.5% Na deoxycholate, 0.1% SDS, 1 mM EDTA). Proteins were resolved by SDS-PAGE and then transferred onto PVDF membranes (Millipore). All buffers contained a cocktail of protease inhibitors (Boehringer, Ingelheim am Rhein, Germany). Membranes were developed using the enhanced chemiluminescence (ECL westar, Cynagen, Bologna, Italy). Bands were analyzed by chemiluminescence imaging system, Alliance 2.7 (UVITEC, Cambridge, UK) and quantified by the software Alliance V_1607.

The following primary antibodies were used: rabbit α-PTTG1 (Abcam), mouse α-MMP-2 (Thermo Fisher, Waltham, MA, USA), mouse α-HA (BioLegend, San Diego, CA, USA), mouse α-tubulin (Sigma, St. Louis, MO, USA), mouse α-actin monoclonal antibody C-40 (Sigma), and rabbit α-Sp1 (Santa Cruz, Santa Cruz, CA, USA).

### 4.5. Isolation of Nuclear/Cytoplasmic Fractions

Nuclear and cytoplasmic fractions were prepared as follows: cells, scraped off the plate with PBS, were resuspended in hypotonic lysis buffer (10 mM HEPES pH 7.9, 10 mM KCl, 0.1 mM EDTA, 0.1 mM EGTA) added with protease inhibitors (Boehringer). After resuspension, NP-40 was added to a final concentration of 0.6% and the nuclei are isolated by centrifugation at 300× *g* for 5 min at 4 °C. After removal of the supernatant (i.e., the cytoplasmic extract), nuclei were resuspended in nuclear extract buffer (20 mM HEPES pH 7.9, 25% glycerol, 0.4 M NaCl, 0.1 mM EDTA, 0.1 mM EGTA), sonicated three times for 5 s at 20% amplitude, and then recovered by centrifugation at 15,000× *g* for 5 min at 4 °C.

### 4.6. Invasion Assay

For the invasion assay, the cells (2.5 × 10^4^ cells in serum-free medium) were seeded in the upper well of the Transwell chamber (8-mm pore size; Corning Glass, Corning, New York, NY, USA) that was precoated with 10 mg/mL growth factor-reduced Matrigel (BD Biosciences, Franklin Lakes, NJ, USA). The lower well was filled with 0.8 mL of growth medium containing 10% fetal bovine serum as a chemoattractant. After incubation for 48 h at 37 °C, non-invaded cells on the upper surface of the filter were removed with a cotton swab, and migrated cells on the lower surface of the filter were fixed and stained with a Diff-Quick kit (Fisher, Waltham, MA, USA). Cells were imaged by phase contrast microscopy (Leica Microsystems, Wetzlar, Germany; magnification 10×). Invasiveness is determined by counting cells in five microscopic fields per well, and the extent of invasion is expressed as an average number of cells per microscopic field.

### 4.7. Gelatin Zymography

MMP-2 activity in the supernatant of seminoma cells was measured by gelatin zymography. Samples were loaded on SDS polyacrylamide gel containing 0.1% gelatin. Following electrophoresis, gels were washed three times for 10 min at room temperature in 2.5% Triton X-100 to remove the SDS. After overnight incubation at 37 °C in a zinc- and calcium-chloride-containing buffer at 37 °C allowing gelatin degradation by gelatinases, gels were stained with 0.5% coomassie brilliant blue for 30 min and finally destained in 20% methanol and 10% acetic acid. Gelatinolytic activities were observed as a clear band of digested gelatin on a blue background. Images were acquired with a digital camera (Nikon, Tokyo, Japan), and bands were acquired with Gel Doc 2000 (Biorad, Hercules, CA, USA) and quantified by the software Alliance V_1607 (UVITEC, Cambridge, UK).

### 4.8. Wound-Healing Assay

An in vitro wound-healing assay was used to observe the migration of the seminoma cell lines. Cells were seeded in a 6 cm dish for 24 h until they visibly reached confluence. A pipette tip was used to create a straight scratch on the plate to simulate a wound. The width of the remaining gap was imaged using phase-contrast microscopy (Zeiss Axiovert200, 10× magnification; Zeiss, Oberkochen, Germany) at the indicated time (4 and 24 h).

### 4.9. Co-Localization Analysis

To quantify the compartmentalization of PTTG1 within the nuclei of different cells, we applied a co-localization approach. Co-localization analysis was performed through the Co-localization Threshold plugin available in the open source software ImageJ (NIH). The analysis of fluorescence co-localization was represented graphically in scatterplots, where the intensity of the blue channel is plotted versus the intensity of the green channel for each pixel, as previously reported [39]. A proportional fluorescence intensity of the two probes results in the distribution of points along a straight line, with the slope reflecting the ratio of the fluorescence of the two probes. To quantify the intranuclear compartmentalization of PTTG1, we evaluated the Manders’ co-localization coefficients. In particular, for two probes, denoted as B and G, respectively, the coefficient M2 provides the fraction of G, i.e., the fraction of pixels expressing protein, in compartments containing B, which is DAPI-stained nuclei, according to the following formula:$$M_{2}=\frac{\sum_{i}G_{i,colocal}}{\sum_{i}G_{i}}\;\mathrm{where}\;\left\{\begin{array}{c} G_{i,colocal}=\;G_{i}\;if\;B_{i}>0\\ G_{i,colocal}=\;0\;if\;B_{i}=0 \end{array}\right.$$

Threshold value was automatically identified by applying the Costes method [40] and the coefficient was evaluated only for pixels above threshold (tM2), with values ranging from 1, when the protein is expressed in the whole nucleus, to 0, in case of no expression of the protein.

### 4.10. Analysis of Testicular Tumors from Atlas Database

To analyze PTTG1 mRNA levels we used RNA-seq data expressed as median FPKM (number fragments per kilobase of exon per million reads) levels of PTTG1 in non-seminoma (N-S; *n* = 65) and seminoma (S; *n* = 68) specimens in the Atlas database (https://www.proteinatlas.org/ENSG00000164611-PTTG1/pathology/testis+cancer#). Immunohystochemical analysis of PTTG1 in non-seminoma (N-S; *n* = 4) and seminoma (S; *n* = 7) specimens was performed using CAB008373 antibody. To analyze PBF mRNA levels, we used RNA-seq data expressed as median FPKM (number fragments per kilobase of exon per million reads) levels of PBF in non-seminoma (N-S; *n* = 65) and seminoma (S; *n* = 68) specimens in the Atlas database (https://www.proteinatlas.org/ENSG00000183255-PTTG1IP/pathology/testis+cancer). Immunohystochemical analysis of PBF in non-seminoma (N-S; N = 4) and seminoma (S; N = 7) specimens was performed using CAB034146 antibody. To analyze MMP-2 mRNA levels, we used RNA-seq data expressed as median FPKM levels of MMP-2 in non-seminoma (N-S; *n* = 65) and seminoma (S; *n* = 68) specimens in the Atlas database (https://www.proteinatlas.org/ENSG00000087245-MMP-2/pathology/testis+cancer).

### 4.11. Statistical Analysis

Results were expressed as values of mean ± standard deviation (SD). The statistical test used was paired two-tailed Student *t*-test. A *p* < 0.05 was considered as significant.

The Atlas database was analyzed by unpaired *t*-test with Welch’s correction (that is, do not assume equal SDs). Outliers (only for MMP-2) were calculated by the ROUT method (robust regression followed by outlier identification) using a False Discovery Rate set to 0.1%. The software used was GraphPad Prism 7.04.

## 5. Conclusions

The present study strengthens the role of PTTG1 nuclear localization in promoting invasiveness and metastasis of seminoma cell lines. Analysis of Atlas in vivo data strongly supported these results, revealing an exclusive PTTG1 nuclear localization and a concomitant increase of MMP-2 levels in seminoma compared to non-seminoma tumors. Overall, these data strongly suggest that nuclear PTTG1 may be a prognostic factor in seminomas.

## Figures and Tables

**Figure 1 cancers-13-00212-f001:**
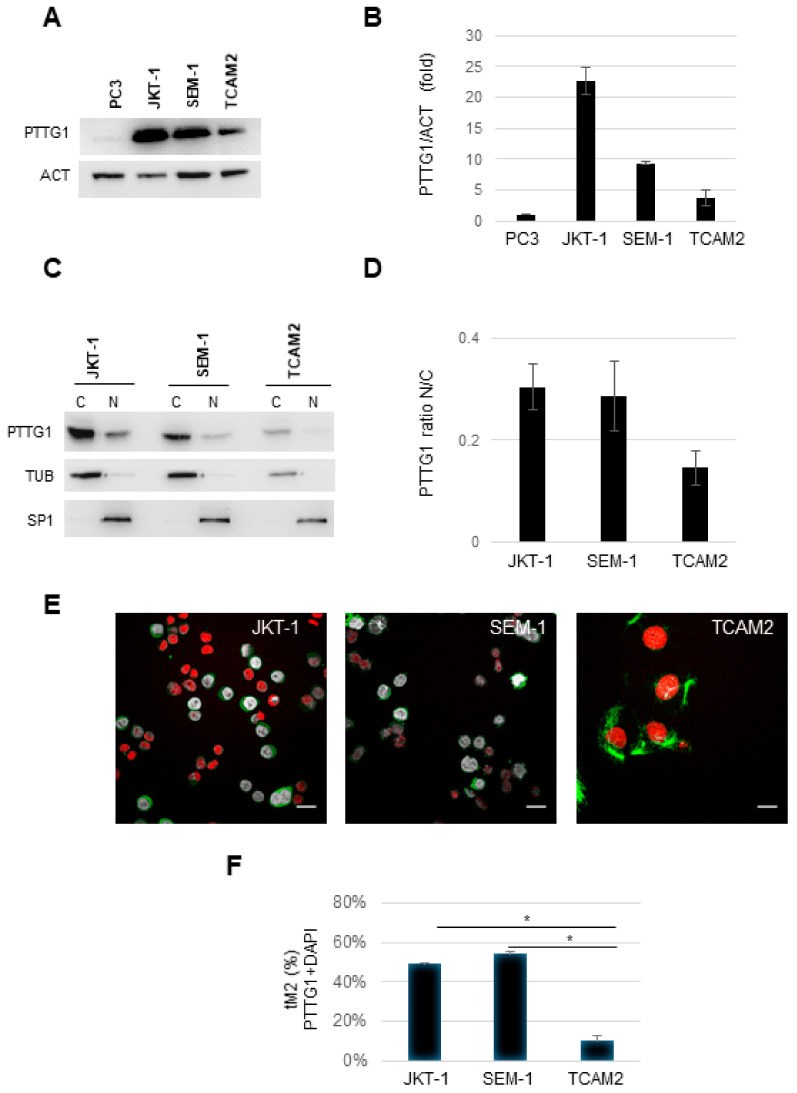
(**A**) Representative Western blot (WB) analysis of the indicated proteins in PC3, JKT-1, SEM-1, and TCAM2 cell lines. (**B**) Histogram shows the ratio of densitometric values of pituitary tumor-transforming gene 1 (PTTG1) to actin (ACT). Mean ± SD of two independent biological replicates is shown (*n* = 2). (**C**) Representative WB analysis of the indicated proteins in JKT-1, SEM-1, and TCAM2 cell lines. Cell lysates are fractionated in cytoplasmic (C) and nuclear (N) compartments. (**D**) Histogram shows the ratio of PTTG1 between nuclear and cytoplasmic fractions (N/C) for each cell line, previously normalized to tubulin (TUB) for cytoplasmic fractions and to specificity protein 1 (SP1) for nuclear fractions. Mean ± SD of two independent biological replicates is shown (*n* = 2). (**E**) Representative pictures of merged confocal immunofluorescence analysis of PTTG1 (green) and nuclei (counterstained with DAPI, here shown in red) in JKT-1, SEM-1, and TCAM2 cells, respectively. The PTTG1-DAPI co-localized pixels are shown in gray. Scale bar: 20 μm. (**F**) Histogram reports the value of tM2 co-localization coefficient of PTTG1-DAPI, expressed in percentage. For each cell line, three fields were counted (*n* = 150 for JKT-1, *n* = 100 for SEM-1, and *n* = 30 for TCAM2; * = *p* < 0.05, two-tailed unpaired *t*-test).

**Figure 2 cancers-13-00212-f002:**
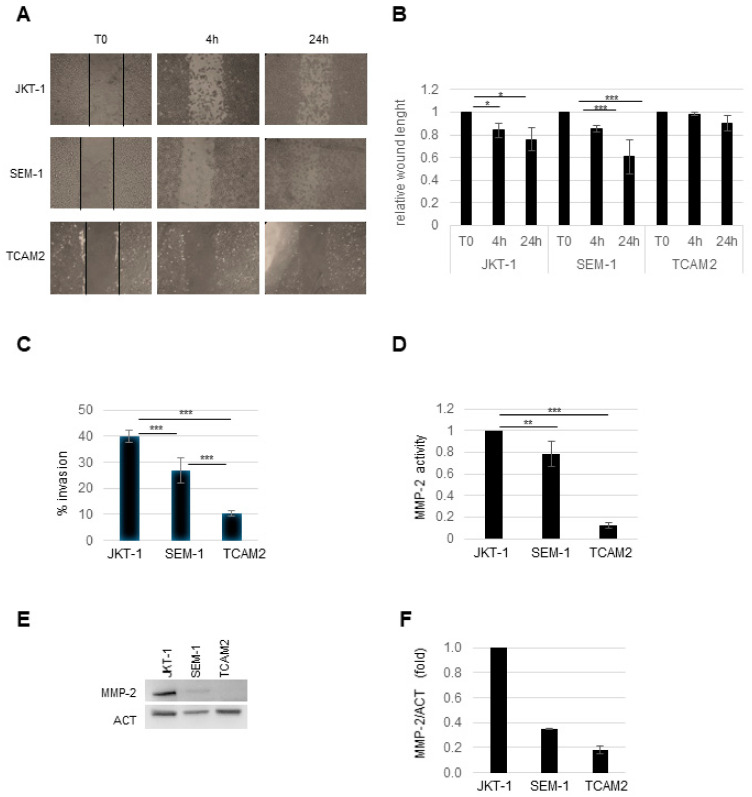
(**A**) Representative pictures of wound-healing assay in JKT-1, SEM-1, and TCAM2 recorded at 4 and 24 h after wounding. Lines show the wound length at time zero. (**B**) Histogram shows the relative wound length in the three cell lines. The length at T0 of each cell line is arbitrarily set to 1 (*n* = 3, * = *p* < 0.05, *** = *p* < 0.001, two-tailed unpaired *t*-test). (**C**) Histogram shows % of cells that invade the matrix compared to the total plated cells in the three cell lines (*n* = 3, *** = *p* < 0.001, two-tailed unpaired *t*-test). (**D**) Histogram shows the matrix-metalloproteinase-2 (MMP-2) activity analyzed by zymography in the three cell lines. MMP-2 activity in JKT-1 was arbitrarily set to 1 (*n* = 3, ** = *p* < 0.01, *** = *p* < 0.001, two-tailed unpaired *t*-test). (**E**) Representative WB analysis of the indicated proteins in JKT-1, SEM-1, and TCAM2 cell lines. (**F**) Histogram shows the ratio of densitometric values of MMP-2 to actin (ACT). Mean ± SD of two independent biological replicates is shown (*n* = 2).

**Figure 3 cancers-13-00212-f003:**
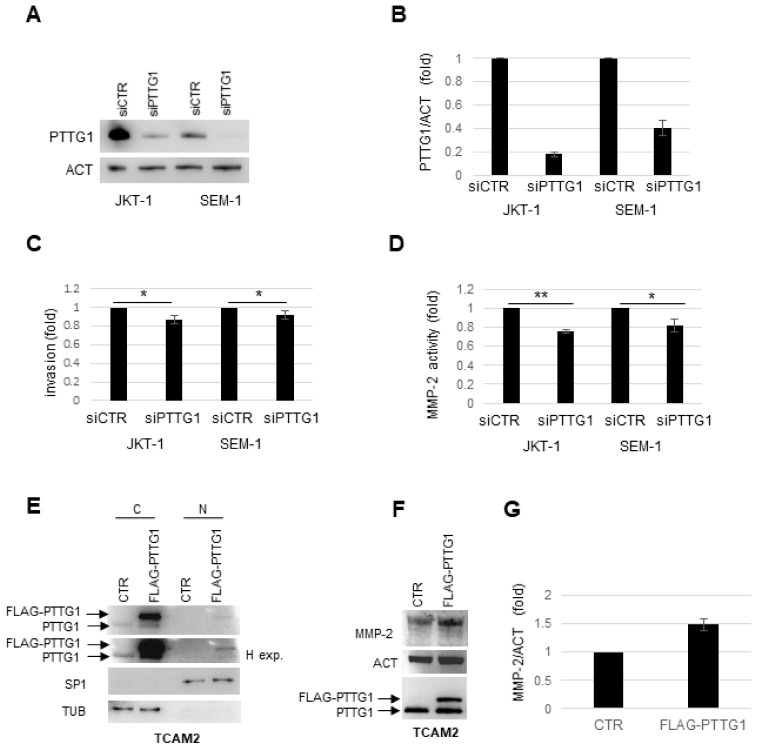
(**A**) Representative WB analysis of the indicated proteins in JKT-1 and SEM-1 cell lines transfected with siCTR or siPTTG1. (**B**) Histogram shows the ratio of densitometric values of PTTG1 to actin (ACT). The ratio PTTG1/ACT from siCTR lane was arbitrarily set to 1. Mean ± SD of two independent biological replicates are shown (*n* = 2). (**C**) Histogram shows matrix cells invasion compared to the total plated cells in the indicated cell lines transfected with siCTR or siPTTG1. The number of invaded cells in siCTR treated cells was arbitrarily set to 1 in each cell line (*n* = 3, * = *p* < 0.05). (**D**) Histogram shows the MMP-2 activity analyzed by zymography in JKT-1 and SEM-1 transfected with siCTR or siPTTG1. MMP-2 activity in siCTR transfected was arbitrarily set to 1 in each cell line (*n*= 3, * = *p* < 0.05, ** = *p* < 0.01, two-tailed unpaired *t*-test). (**E**) WB analysis of the indicated proteins in TCAM2 cells. Cell lysates are fractionated in cytoplasmic (C) and nuclear (N) compartments. Arrows indicate endogenous PTTG1 signal and FLAG-PTTG1 overexpression. H exp. indicates higher exposure time. (**F**) Representative WB analysis of the indicated proteins in TCAM2 cells transfected with CTR or FLAG-PTTG1 plasmids. (**G**) Histogram shows the ratio of densitometric values of MMP-2 to actin (ACT). The ratio MMP-2/ACT from CTR lane was arbitrarily set to 1. Mean ± SD of two independent biological replicates are shown (*n* = 2).

**Figure 4 cancers-13-00212-f004:**
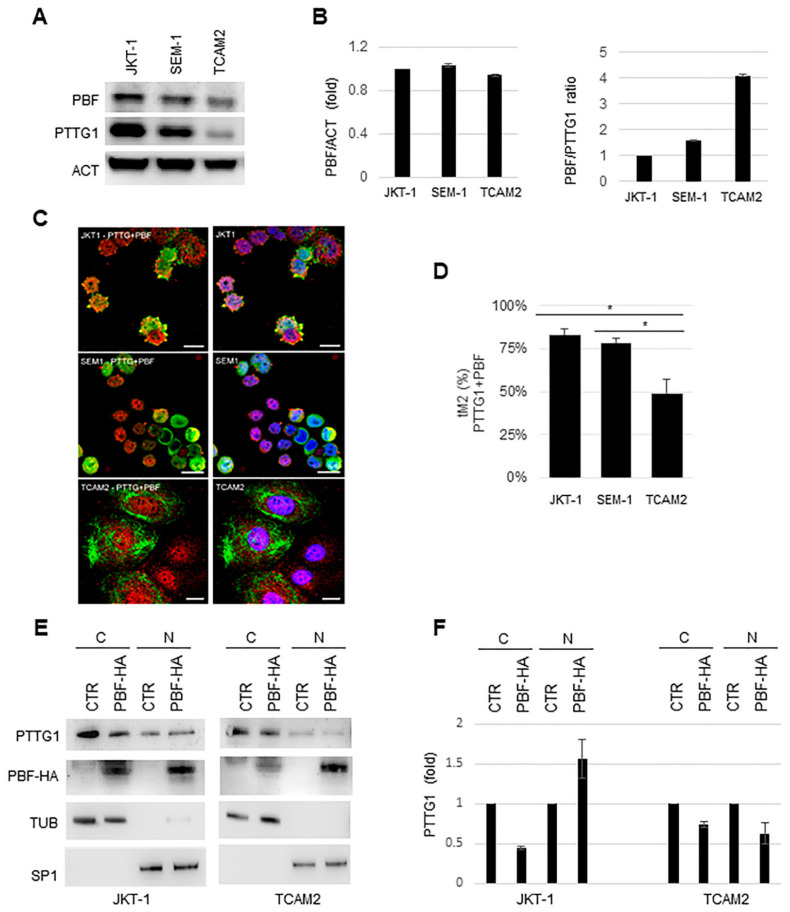
(**A**) Representative WB analysis of the indicated proteins in JKT-1, SEM-1, and TCAM2 cell lines. (**B**) Left panel: histogram shows the ratio of densitometric values of PTTG-binding factor (PBF) to actin (ACT). The ratio PBF/ACT from JKT-1 lane was arbitrarily set to 1. Mean ± SD of two independent biological replicates is shown (*n* = 2). Right panel: histogram shows the ratio between PBF and PTTG1 (PBF/PTTG1) for each cell line, previously normalized to actin (ACT). The ratio PBF/PTTG1 from JKT-1 lane was arbitrarily set to 1. Mean ± SD of two independent biological replicates is shown (*n* = 2). (**C**) Representative pictures of merged confocal immunofluorescence analysis of PTTG1 (green) and PBF (red) (left panel), in JKT-1, SEM-1, and TCAM2 cells, respectively. Nuclei, counterstained with DAPI (blue), are shown in the right panel. Scale bar: 20 μm. (**D**) Histogram shows the value of tM2 co-localization coefficient of PTTG1-PBF, expressed in percentage. For each cell line, 3 fields were counted (*n* = 60 for JKT-1, *n* = 120 for SEM-1, and *n* = 30 for TCAM2; * = *p* < 0.05, two-tailed unpaired *t*-test). (**E**) Representative WB analysis of the indicated proteins in JKT-1 and TCAM2 cell lines transfected with CTR or PBF-HA plasmids. Cell lysates were fractionated in cytoplasmic (C) and nuclear (N) compartments. (**F**) Histogram shows the densitometric values of PTTG1 previously normalized to tubulin (TUB) for cytoplasmic fractions and to specificity protein 1 (SP1) for nuclear fractions. The ratio of PTTG1 relative value from CTR lane of each cell compartments (C or N) in the different cell lines was arbitrarily set to 1. Mean ± SD of two independent biological replicates is shown (*n* = 2).

**Figure 5 cancers-13-00212-f005:**
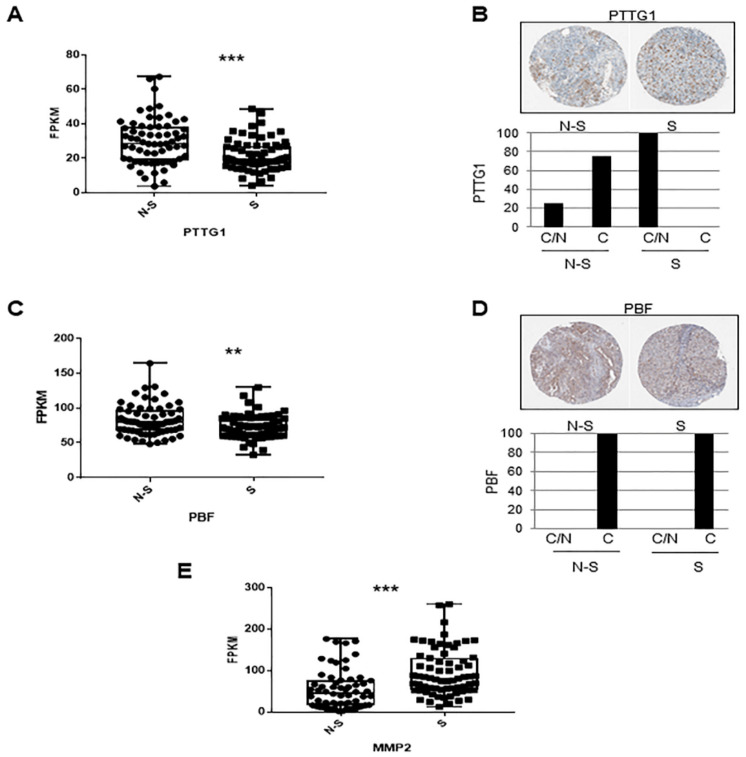
(**A**) Box plot of RNA-seq data of PTTG1 in non-seminoma (N-S; *n* = 65) and seminoma (S; n*N* = 68) specimens in the Atlas database. *** = *p* value < 0.0001. (**B**) Immunohystochemical analysis of PTTG1 in non-seminoma (N-S; *n* = 4) and seminoma (S; *n* = 7) specimens. Upper panel show a representative samples image in N-S and in S (https://www.proteinatlas.org/ENSG00000164611-PTTG1/pathology/testis+cancer#img). Histogram shows the percentage of PTTG1 in the cytoplasmic/membranous/nuclear localization (C/N) and cytoplasmic/membranous (C) compartments. (**C**) Box plot of RNA-seq data of PBF in non-seminoma (N-S; *n* = 65) and seminoma (S; *n* = 68) specimens in the Atlas database. ** = *p* value < 0.01. (**D**) Immunohystochemical analysis of PBF in non-seminoma (N-S; *n* = 4) and seminoma (S; *n* = 7). Upper panel show representative sample images in N-S and in S (https://www.proteinatlas.org/ENSG00000183255-PTTG1IP/pathology/testis+cancer#img). Histogram shows the percentage of PBF in the cytoplasmic/membranous/nuclear localization (C/N) and cytoplasmic/membranous (C) compartments. (**E**) Box plot of RNA-seq data of MMP-2 in non-seminoma (N-S; In = 65) and seminoma (S; *n* = 68) specimens in the Atlas testicular tumor database. *** = *p* value < 0.0001.

## Data Availability

The data presented in this study are available on request from the corresponding author.

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
