# Peer review of "Nuclear Localization of PTTG1 Promotes Migration and Invasion of Seminoma Tumor through Activation of MMP-2"

_cancers, 2021, doi:10.3390/cancers13020212_

Round 1

Reviewer 1 Report

In their manuscript Teveroni et al. studied in vitro the role of PTTG1 on the invasive/migratory properties of seminoma and showed that the nuclear localization of PTTG1 as well as its transcriptional target Matrix-Metalloproteinase-2 (MMP-2) might play an important role. Together with data on PTTG1 expression and localization in seminoma which were acquired from the Atlas database the authors conclude that PTTG1 might not only play an important role for invasiveness and metastasis of seminoma, but could also serve as a predictive factor in seminomas.

Despite of some minor grammatical and typing errors the text is well written and the authors' hypothesis is clear. The set of experiments chosen to explore the interplay of PTTG1 localization, MMP-2 activity and seminoma cell migration is logical and well-chosen. The presented work is original, offers some new insight into the role of PTTG1 in seminoma and may thus be of interest for the readers of Cancers.

I have only a few questions and remarks that need to be answered/addressed before publication:

  • Even after upregulation of PTTG1 in Tecam-2 cells, nuclear localization and MMP-2-induced migration were not increased accordingly. So, the authors concluded that additional factors might play a role. After excluding a role for PBF, the authors then hypothesized that posttranscriptional modifications, such as PTTG1-phosphorylation might be responsible. But instead of investigating this interesting hypothesis, they finished their study by declaring that this may be something for future investigations. I had expected to see that the authors addressed this idea experimentally. It would have helped to explain the at least partially contradictory findings on TECAM-2 cells as compared to the other two sweminoma cell lines.
  • The quality of the Western Blots seems to be fine at a first sight. But looking at the supplementary files showing the original blots it becomes apparent that there were many unspecific bands in each of the blots. As no size-ladder is included in the original blots, I wonder how the authors could be sure to have identified the right bands on these blots?
  • What is the reason for taking tubulin as a housekeeping protein for standardization in figure 3e and figure S2, while ß-actin was used in all the other experiments?
  • Figure1: PC3 cells were taken as a “positive control” for PTTG1-overexpressing cancer cells. However, the Western Blot shows hardly any expression in these cells (figure 1a)?

Reviewer 2 Report

  1. In line 353, the names of cell lines used in this study should be listed in Materials and methods section.
  2. In line 182, spelling should be checked (e.g. correct "than" to "then")
  3. In line 152 to 154, author claimed that MMP-2 expression level showed a similar trend compared to its activity in Figure 2DEF. But, MMP2 expression was detected at a very low level in SEM-1 cell line (Figure 2E). How can you explain the co-relationship between MMP2 expression level and activity in this cell line?
  4. Cell lines marked in the Figure 4E.
  5. The method using the Atlas database should be described in detail in the materials and methods section, not the result section.
